# Microstructure and Creep Properties of Ni-Base Superalloy IN718 Built up by Selective Laser Melting in a Vacuum Environment

**Toshiki Nagahari [1],\*, Taigi Nagoya [1], Koji Kakehi [1],\*, Naoko Sato [2]**  **and Shizuka Nakano [2]**

[1] Department of Mechanical Systems Engineering, Tokyo Metropolitan University, 1-1 Minami-osawa, Hachioji-shi, Tokyo 192-0397, Japan; nagoya-taigi@ed.tmu.ac.jp

[2] National Institute of Advanced Industrial Science and Technology, 1-2-1 Namiki, Tsukuba-shi, Ibaraki 305-8564, Japan; n.sato@aist.go.jp (N.S.); shizuka.nakano@aist.go.jp (S.N.)

\* Correspondence: nagahari-toshiki@ed.tmu.ac.jp (T.N.); kkakehi@tmu.ac.jp (K.K.); Tel.: +80-42-677-2712 (T.N.)

**Abstract:** Selective laser melting (SLM) in a vacuum environment is a relatively new process. Although the material is expected to undergo a gradual heat change, which will influence the microstructure and creep properties of IN718, little research has been conducted to date. Here, we compared specimens built in vacuum (V-SLM) to those built in an Ar atmosphere (Ar-SLM). We investigated the microstructure and creep properties of V-SLM, and compared the V-SLM specimen to a conventional Ar-SLM specimen. The V-SLM specimen had a larger-grained texture, and the quantity of the δ phase was much lower. In addition, the V-SLM specimen had lower amounts of aluminum and titanium oxides, which improved the stability of the γ″ phase. Thus, the V-SLM specimen showed better creep life than the Ar-SLM, due to prevention of brittle fractures along the interdendritic regions.

**Keywords:** selective laser melting; vacuum; Inconel 718; creep

## 1. Introduction

Nickel-based superalloys possess excellent properties, such as corrosion resistance, strength and toughness at high temperatures [1,2]. Compared to other superalloys, Inconel 718 also has good weldability and structural stability at temperatures up to 650 °C. Thus, IN718 superalloy has been widely used in gas turbine disks, aircraft engines, and nuclear reactors [3,4]. However, conventional casting techniques are limited with regard to component complexity. The ability to produce geometries of high complexity is needed to reduce component weight and material consumption [5,6]. Additive manufacturing (AM) technologies are highly attractive. AM processes can produce complex three-dimensional (3-D) structures, as these techniques employ slice data from CAD models [7]. The selective laser melting (SLM) method is an AM process that offers a large degree of design flexibility. In addition, the SLM process has many advantages in manufacturing components, such as extraction rate and dimensional accuracy [8].

However, the use of IN718 in the SLM process involves a large amount of Laves phases, due to microsegregation resulting from a rapid solidification rate and strong thermal gradient. Laves phases are harmful to IN718 as they cause brittleness [4,9,10]. Moreover, the Laves phase promotes δ phase precipitation during heat treatment. Formation of the δ phase requires 6–8% Nb content [8]. Although it is difficult to form the δ phase in a matrix due to the lack of Nb, it is easy to form this phase near a Laves phase. The δ phase precipitates incoherently in the γ matrix, and its morphology is acicular. Thus, the δ phase can initiate cracking at the interface. In addition, δ phases line up at interdendritic

regions. This is very bad for the mechanical properties of the material [3]. Therefore, it is important to control the distribution and quantity of the δ phase.

Kuo et al. [9] conducted several heat treatments and hot isostatic pressing (HIP) processes for IN718 produced by the SLM process. They found that creep properties were improved by new heat treatment and HIP processes. Nevertheless, the creep properties of SLM-fabricated IN718 were not comparable to those of a wrought specimen [11]. The microstructure and properties of metal are determined by its heat history during the manufacturing process. Thus, further investigation of the SLM process is necessary. Moussaoui et al. [12] examined the effects of process parameters such as laser power, scanning speed, hatch spacing, etc., on the mechanical properties of IN718. They found that the tensile properties of SLM-built IN718 were barely altered by variation in the process parameters. As the process parameters are limited in the manufacture of dense samples, heat history during the SLM process was also hardly changed by the process parameters. Thus, investigation of the SLM process from another perspective is needed.

In this study, we fabricated IN718 using the SLM process in a vacuum. By introducing the vacuum environment, we expected the material to undergo a more gradual heat change, and that this would influence the microstructure and creep properties of IN718. In addition, as it is not necessary to pre-heat samples in such an EBM (Electron Beam Melting) process, we expected that dimension accuracy would be good and the vacuum environment would promote the elimination of contaminants, especially oxygen. However, there has been little research on the SLM process in a vacuum. Thus, the objective of this study was to reveal the microstructure of IN718 built by the vacuum SLM process and determine its effect on creep properties.

## 2. Materials and Methods

The experiments were performed on an SLM system called the RaFaEl prototype (Aspect, Inc., Tokyo, Japan). This is a vacuum SLM system, in which the degree of vacuum can be raised up to $1.0 \times 10^{-2}$ Pa. Process parameters are listed in Table 1; the chemical composition of the IN718 powder used in the V-SLM process is presented in Table 2 (powder particle size is under 63 μm). The powder used in the Ar-SLM process was EOS NickelAlloy IN718 [3,11]. The RaFaEl prototype can build samples in a vacuum environment, and such specimens are called vacuum-SLM (V-SLM). IN718 specimens built with a 3-D printer (EOSINT M280; EOS, Robert-Stirling-Ring 1, 82152, Krailling, Bavaria, Germany) were also prepared for comparison (here, called Ar-SLM specimens). Standard heat treatment was conducted for both V-SLM and Ar-SLM specimens. The treatment protocol begins with a solution treatment at 980 °C for 1 h, followed by air cooling to room temperature. Subsequently, there is a two-step aging treatment: first, specimens are held at 718 °C for 8 h, then furnace-cooled to 621 °C; second, specimens are held at 621 °C for 10 h, then air-cooled to room temperature. The heat-treated specimens will be called HT specimens, in contrast with as-built specimens.

**Table 1.** Process parameters of V-SLM (vacuum selective laser melting).

| Power (W) | Scanning Speed (mm/s) | Hatch Spacing (mm) | Thickness (mm) |
|---|---|---|---|
| 280 | 450 | 0.03 | 0.2 |

**Table 2.** Chemical composition of IN718 powder used in the V-SLM process (mass %).

| Ni | Cr | Nb | Mo | Ti | Al | Co | Cu | C | Si | Mn | O | Fe |
|---|---|---|---|---|---|---|---|---|---|---|---|---|
| 52.73 | 18.92 | 5.21 | 3.02 | 0.90 | 0.62 | 0.02 | 0.04 | 0.06 | 0.19 | 0.20 | 160 ppm | Bal. |

After heat treatment, specimens for creep tests were cut from the cube using a spark cutter. The gauge dimension of each specimen was $19.6 \times 2.8 \times 3.0$ mm$^3$. The creep test parameters were set as 650 °C/550 MPa, and a creep test was conducted for each specimen. The microstructures were observed by optical microscope (OM; Olympus Corp. Tokyo, Japan), scanning electron microscope (SEM; Hitachi,

Ltd., Japan), and transmission electron microscope (TEM; JEOL Ltd., Tokyo, Japan). Inverse pole and pole figures were calculated from the orientation measurements by electron backscatter diffraction (EBSD; Oxford Instruments, Oxfordshire, UK). The image processing software Image J bundled with 64-bit Java 1.8.0_112 (American National Institutes of Health. Bethesda, MD, USA) was used for measurements. The oxygen level was analyzed using the inert gas fusion method with a Bruker G8 Galileo (Bruker Co., Billerica, MA, USA).

## 3. Results

### 3.1. Microstructure Observation

#### 3.1.1. As-Built Specimens

Figure 1 shows the microstructure of the as-built specimens. As may be seen in the figure, molten pools can be observed in the Ar-SLM specimen (Figure 1a). Fine dendrites could be identified within the molten pools, with various orientations that generally tended toward the top center of the molten pool (Figure 1b). In contrast, a layered structure was identified in the V-SLM specimen (Figure 1c). The dendrite structure was divided into two regions. One was oriented to the building direction, and the other was skewed toward the building direction (Figure 1d).

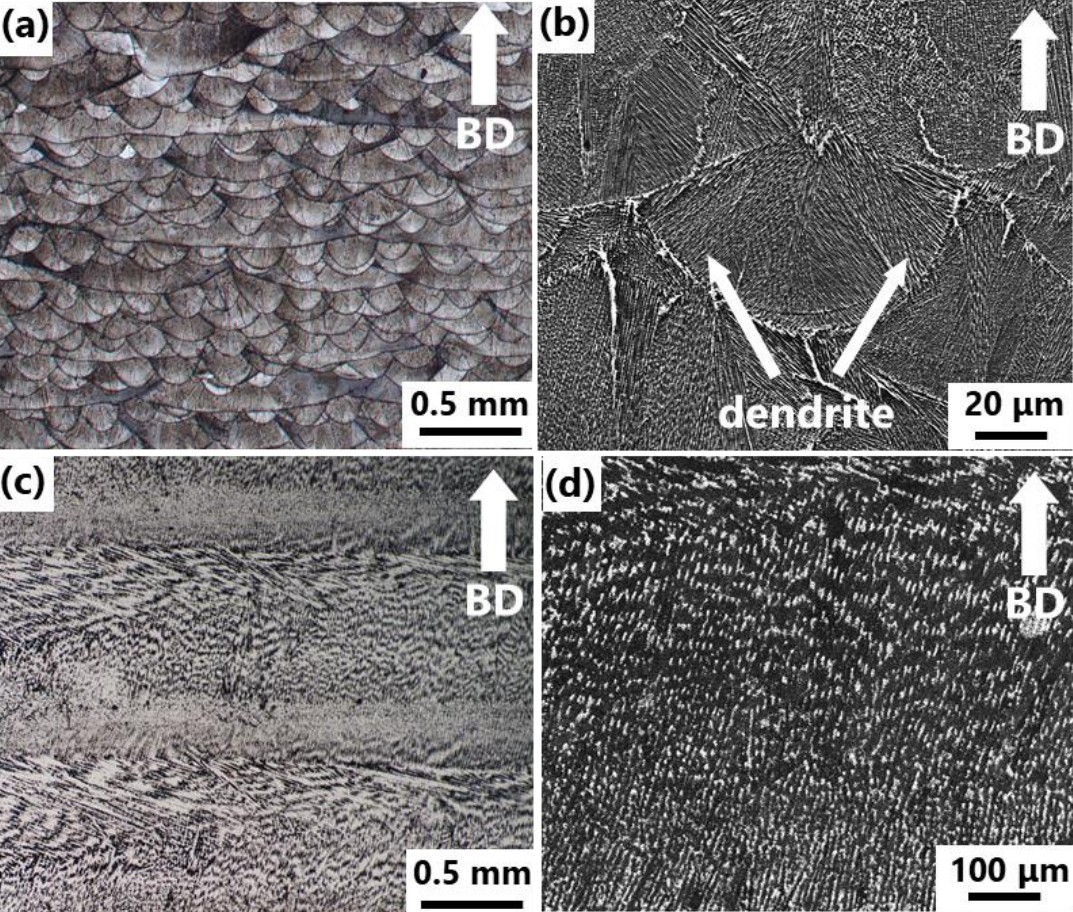

**Figure 1.** Microstructures of the as-built specimens: (**a,b**) Ar-SLM and (**c,d**) V-SLM.

As shown in Figure 2, the dendrite structure of the V-SLM specimen was coarser than that of the Ar-SLM specimen. Dendrite arm spacing in the Ar-SLM (Figure 2a) and V-SLM (Figure 2c) specimens was measured as 0.5 µm and 10 µm, respectively. In the Ar-SLM specimen, the Laves phase precipitated along interdendritic regions, and the size of Laves phase ranged from 50–100 nm (Figure 2b). In the V-SLM specimen, however, the size of the Laves phase was much bigger, and there was a larger distance between dendrites. In addition, the aspect ratio of the dendritic structures was larger and longer, with a length of 5–10 µm.

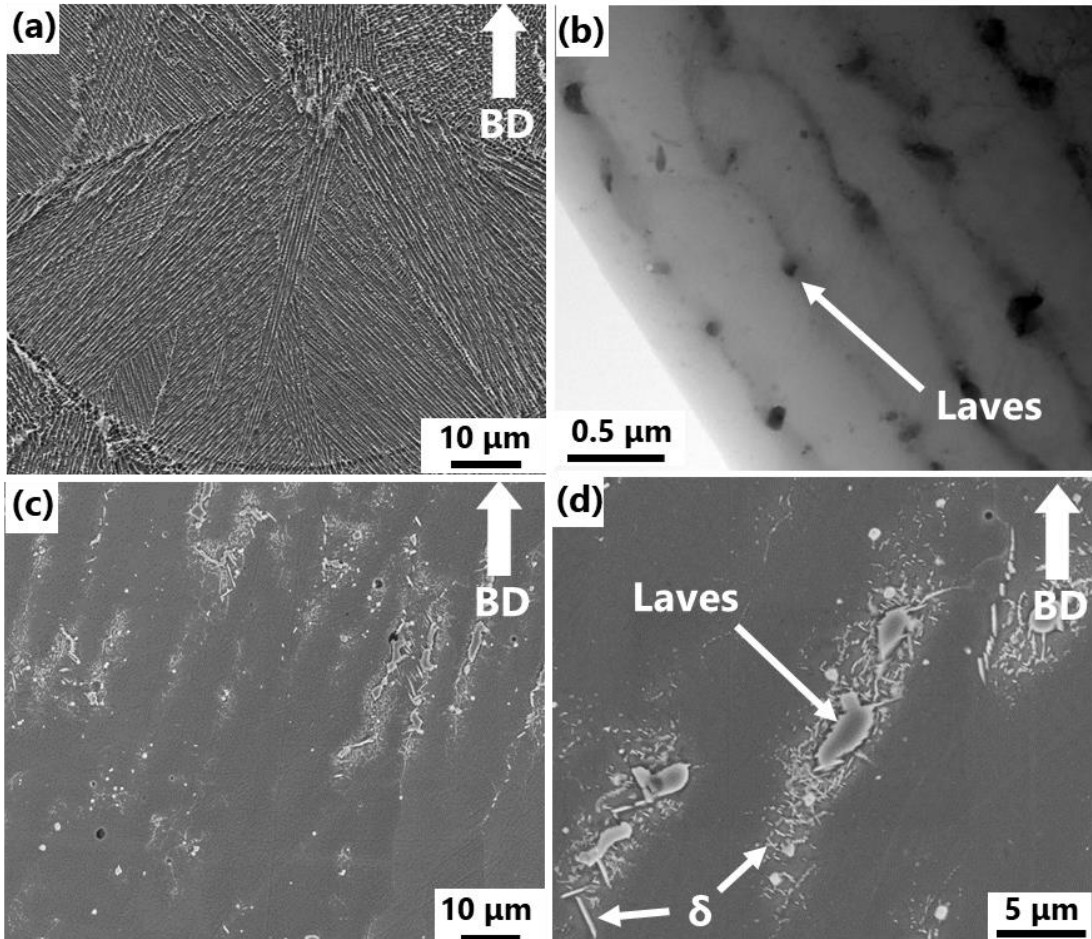

**Figure 2.** Microstructure of the as-built specimens with higher magnification: (**a**,**b**) Ar-SLM [11] and (**c**,**d**) V-SLM. ©2017 Elsevier.

Figure 3 shows the inverse pole figures and pole figures of the as-built specimens, and Table 3 shows the grain size of both specimens. The vertical direction is parallel to the building direction. The Ar-SLM specimen had a mixed-grain texture, which included columnar grains and fine grains (Figure 3a). The columnar grains resulted from epitaxial growth in the center of molten pools, while the fine grains resulted from the material's complex heat history, due to the overlap of laser scanning at the edges of the molten pools. The crystal orientation of the Ar-SLM specimen was varied (Figure 3c), while the V-SLM specimen had larger grains (Figure 3b) with a preferable orientation (Figure 3d).

**Table 3.** Grain size along the vertical and horizontal directions in the Ar-SLM and V-SLM specimens (µm).

| Specimen | Vertical | Horizontal |
|---|---|---|
| Ar-SLM | 80 | 20 |
| V-SLM | 390 | 100 |

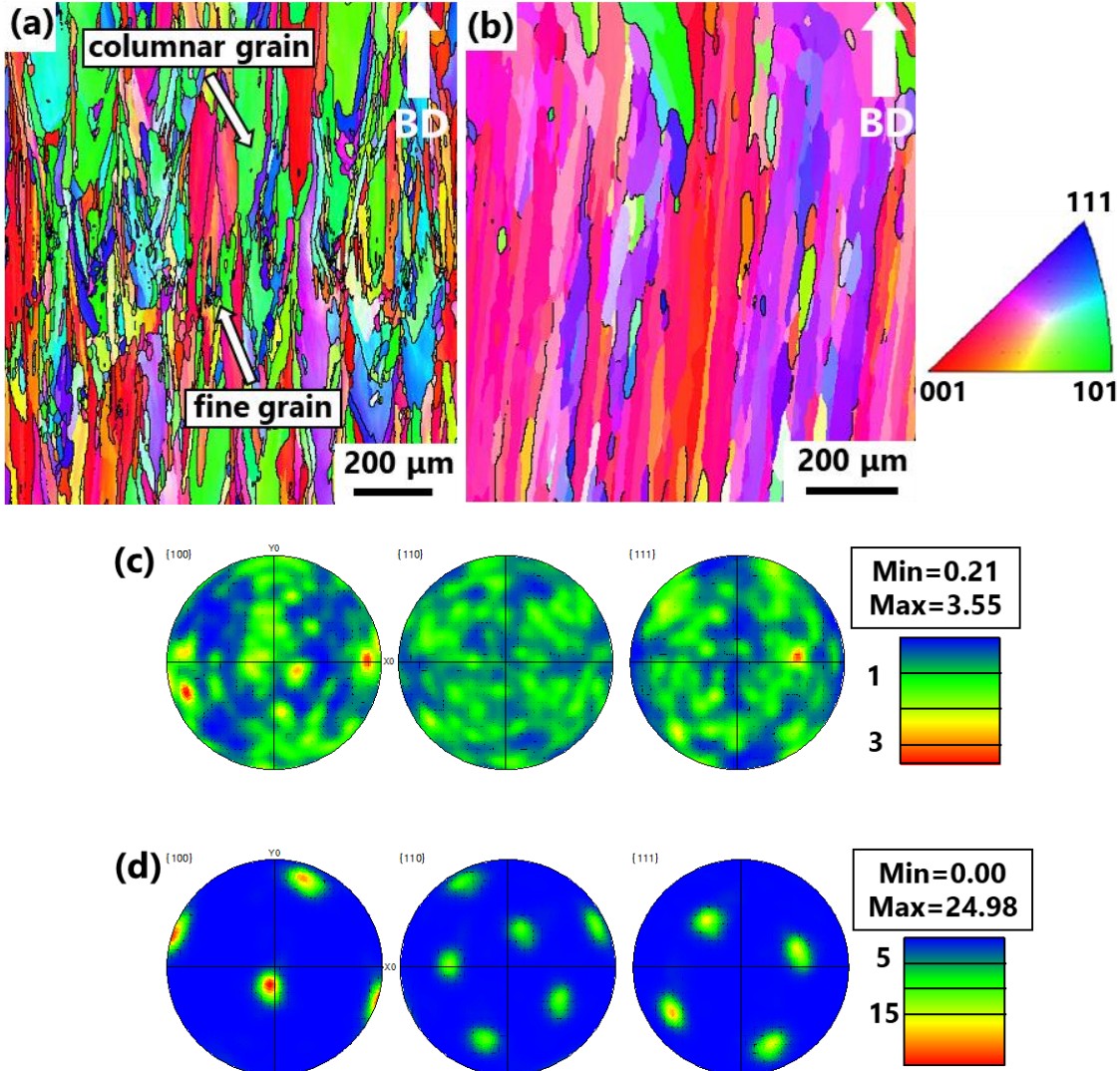

**Figure 3.** Inverse pole figure (IPF) maps of the as-built specimens: (**a**) Ar-SLM and (**b**) V-SLM. Pole figure of the as-built specimens: (**c**) Ar-SLM and (**d**) V-SLM.

3.1.2. HT Specimens

Figure 4 shows the microstructures of the HT specimens. In the Ar-SLM specimen, no molten pools or dendritic structures were observed, and the microstructure was very fine (Figure 4a). In addition, many δ phases were observed along the interdendritic regions and grain boundaries (Figure 4b). For the V-SLM specimen, microstructure of the HT specimen (Figure 4c) was similar to that of the as-built specimen (Figure 2c). In fact, a coarse Laves phase precipitated intermittently along the interdendritic regions, and a δ phase formed around the Laves phase. The only difference was the size of the δ phase. Compared to the δ phases in the as-built specimen (Figure 2d), those in the HT specimen (Figure 4d) were substantially larger. As the δ phase evolved, consuming Nb from the coarse Laves phase during heat treatment, it grew. Table 4 shows the average values of EDS (Energy Dispersive X-Ray Spectroscopy) analysis at Laves phases for the as-built and HT specimens produced by V-SLM. The Nb content of the HT specimen was a little lower than that of the as-built specimen, suggesting the Nb in the Laves phase was consumed by the δ phase during heat treatment. Table 5 shows the number density and volume fraction of the δ phase in the HT specimens. While the V-SLM specimen had less δ phase than the Ar-SLM specimen, the size of the δ phase in the V-SLM specimen was larger. However, the quantity of the δ phase was much lower than that of the Ar-SLM specimen.

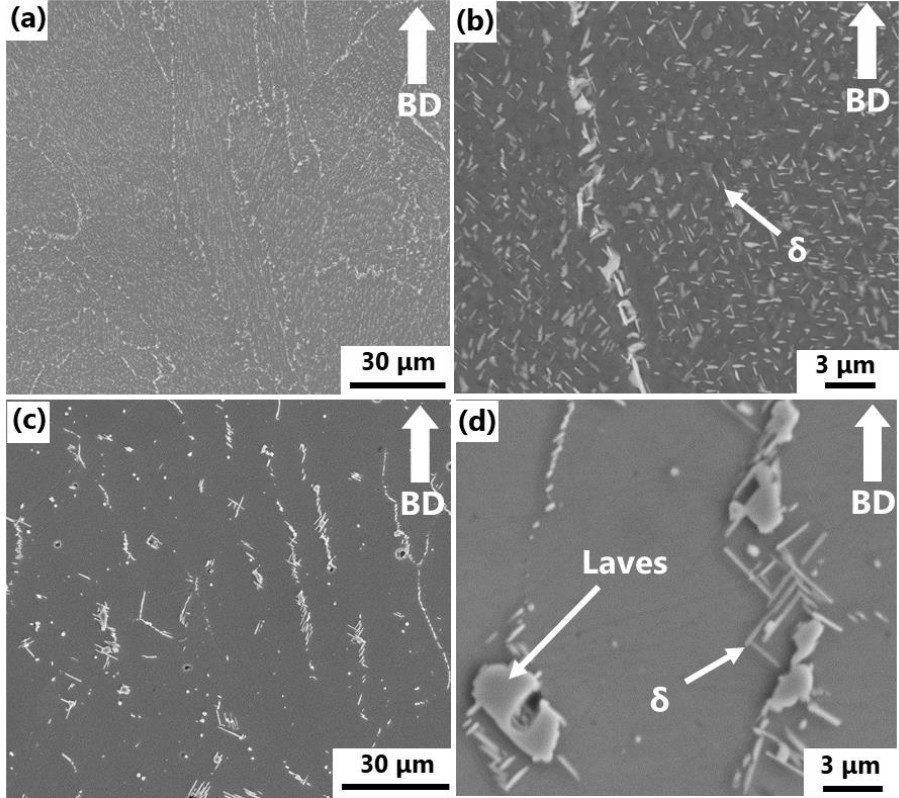

**Figure 4.** Microstructure of the heat-treated (HT) specimens: (**a**,**b**) for the Ar-SLM and (**c**,**d**) for the V-SLM.

**Table 4.** Average values of EDS analysis at Laves phases for V-SLM specimens (mass %).

| HEADING | Ni | Nb | Cr | Fe | Mo | Ti | Al |
|---|---|---|---|---|---|---|---|
| As-Built | 35.6 | 29.8 | 12.8 | 12.8 | 8.0 | 0.7 | 0.2 |
| HT | 39.8 | 27.1 | 11.6 | 11.6 | 8.4 | 1.2 | 0.3 |

**Table 5.** Number density and volume fraction of δ phase in HT specimens.

| Specimen | Number Density (Unit/mm$^2$) | Volume Fraction (%) |
|---|---|---|
| Ar-SLM | $6.6 \times 10^5$ | 8.1 |
| V-SLM | $2.5 \times 10^4$ | 3.4 |

*3.2. Analysis of Oxygen Content*

Table 6 shows the results of oxygen content analysis for the IN718 powder and the SLM-fabricated specimens. For the Ar-SLM specimen, a slight change in oxygen content was confirmed relative to the powder. However, for the V-SLM specimen, the oxygen content of SLM-fabricated specimen was half that of the powder.

**Table 6.** Oxygen content of the powder and the SLM-fabricated specimens (ppm).

| Specimen | Powder | Specimen |
|---|---|---|
| Ar-SLM | 190 | 141 |
| V-SLM | 160 | 78 |

Figure 5 shows EDS mapping of the Ar-SLM HT specimen, which indicates the existence of small Al/Ti-rich oxide particles due to the high oxygen content in Ar-SLM specimen. To compare oxide quantity, backscatter detector (BSE) images were obtained for Ar-SLM and V-SLM specimens (Figure 6). In the Ar-SLM specimen, high-density oxide content was observed, especially near the δ phase (i.e., the interdendritic region (Figure 6a)). Meanwhile, a small amount of oxides was observed, and some existed in the Laves phase. In addition, we calculated the average number density and volume fractions for oxides (Table 7) using 5 SEM images for each specimen. Table 7 shows that the V-SLM specimen had less oxide than the Ar-SLM specimen. This was due to the lower amount of oxygen in the V-SLM specimen. Table 7 includes theoretical values of volume fractions. These values were calculated based on the hypothesis that all oxygen was consumed by aluminum oxides. Theoretical values were close to measured values, confirming the oxide measurement results. The expression used to calculate the theoretical values was as follows:

$$V_O = \frac{\frac{M_O}{d_O}}{\frac{M_M}{d_M}} = \frac{M_O}{M_M} \cdot \frac{d_M}{d_O} \tag{1}$$

where $V_O$ is a theoretical volume fraction, $M_O$ and $M_M$ are the weight of oxygen and metal, respectively, and $d_M$ and $d_O$ are the density of oxygen and metal, respectively.

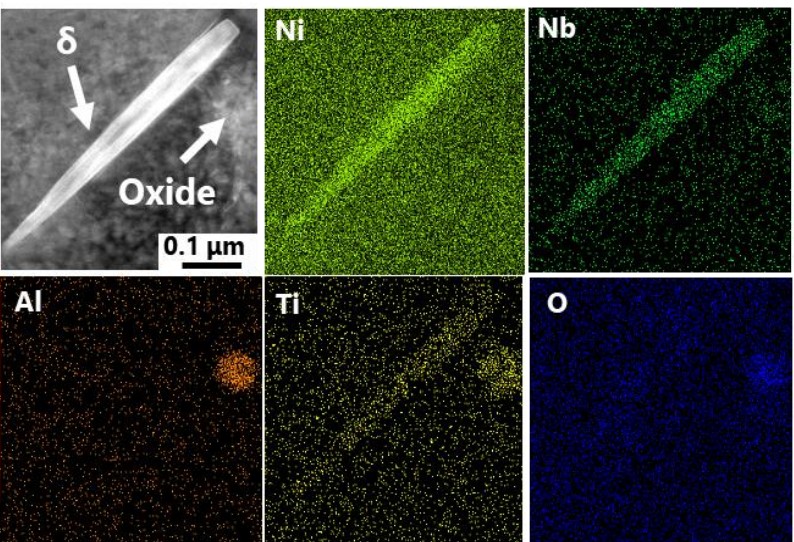

**Figure 5.** EDS mapping of Ar-SLM HT specimen.

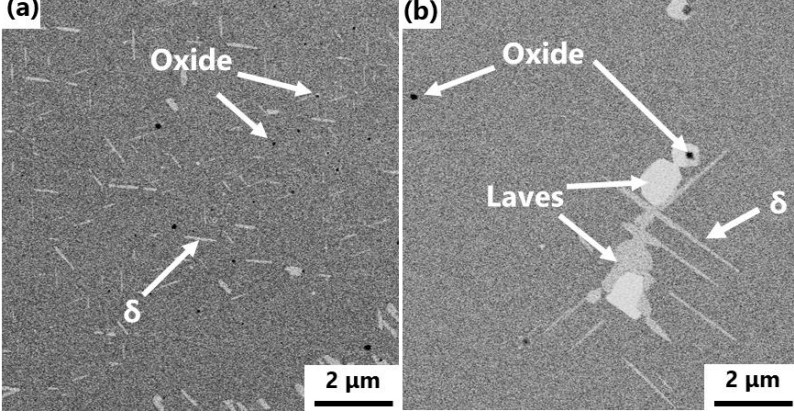

**Figure 6.** BSE (backscatter detector) images of HT specimens: (**a**) Ar-SLM, and (**b**) V-SLM specimens.

**Table 7.** Number density and volume fraction of oxides in SLM-fabricated specimens.

| Specimen | | Number Density (Unit/mm$^2$) | Volume Fraction (10$^{-2}$%) | Theoretical Volume Fraction (10$^{-2}$%) |
|---|---|---|---|---|
| Ar-SLM | As-Built | $2.18 \times 10^5$ | 4.50 | 6.18 |
| | HT | $3.01 \times 10^5$ | 5.88 | - |
| V-SLM | As-Built | $2.70 \times 10^5$ | 1.24 | 3.44 |
| | HT | $1.72 \times 10^5$ | 1.68 | - |

### 3.3. Creep Properties

Figure 7a presents the creep strain–time curve and Figure 7b presents the creep strain rate–time curve for HT specimens under 650 °C/550 MPa. The Ar-SLM specimen exhibited a creep life of 134 h, while the V-SLM specimen exhibited a creep life 2.7 times higher, 359 h. Moreover, the V-SLM exhibited a lower creep strain rate and a larger rupture strain compared to the Ar-SLM. Figure 8 shows rupture surfaces after the creep test. As may be seen, the Ar-SLM specimen showed brittle fractures due to fracturing along interdendritic regions. Meanwhile, the V-SLM specimen experienced ductile fracture. Figure 9 shows the microstructures along the loading direction after the creep test. The Ar-SLM specimen had large cracks (Figure 9a) parallel to the δ-phase distribution (Figure 9c). However, the V-SLM specimen had few large cracks. Instead, in this specimen, there were some voids and microcracks in the Laves phase.

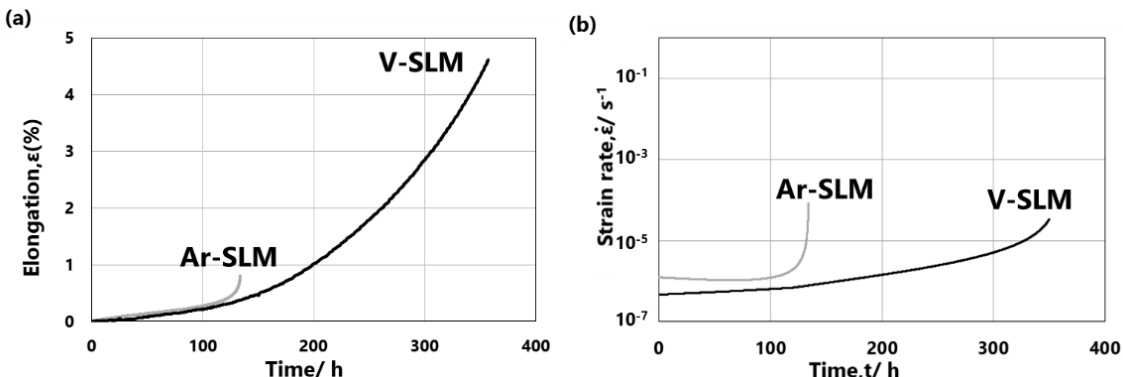

**Figure 7.** Creep curves of HT specimens at 650 °C under 550 MPa: (**a**) creep strain–time curves and (**b**) creep strain rate–time curves.

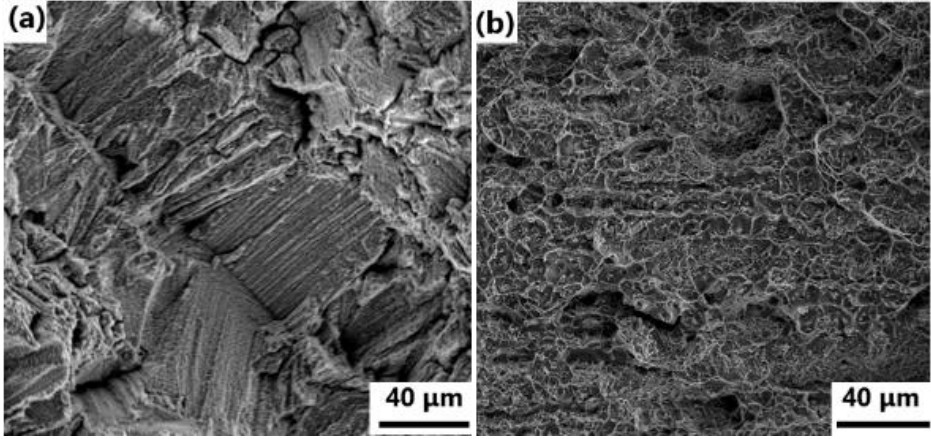

**Figure 8.** Rupture surfaces of: (**a**) Ar-SLM [11], and (**b**) V-SLM specimens, after creep tests at 650 °C under 550 MPa. ©2017 Elsevier.

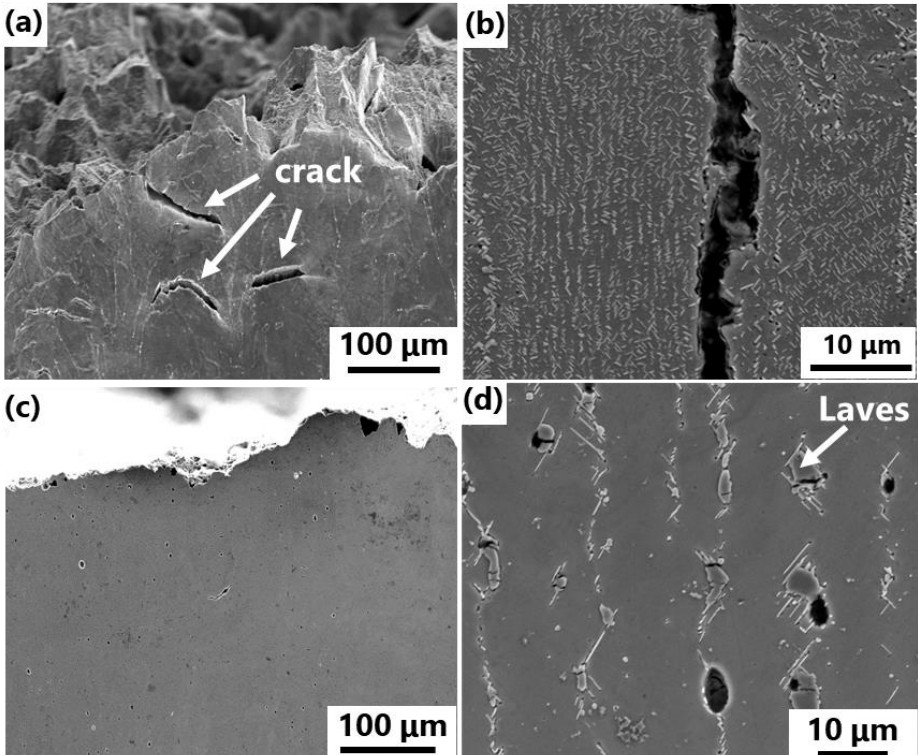

**Figure 9.** Microstructure after creep test: (**a**,**b**) Ar-SLM; and (**c**,**d**) V-SLM specimens.

## 4. Discussion

### 4.1. Characteristics of δ and Laves Phases and Their Effects on Fracture Mechanism

The characteristics of the δ phase in the HT specimens differed between the V-SLM and Ar-SLM specimens. In the Ar-SLM specimen, a lot of fine δ phases precipitated uniformly along the interdendritic regions, and the spacing between phases was small (Figure 4a,c). Meanwhile, in the V-SLM specimen, δ phases precipitated intensively around coarse Laves phases. This was due to the difference in Nb segregation in the as-built condition. Nb segregated along the interdendritic regions during the solidification process [13]. In addition, a high cooling rate led to the precipitation of fine Laves phases, while a low cooling rate led to the precipitation of coarse Laves phases [13].

In this study, many fine Laves phases precipitated in the Ar-SLM specimen, while a small amount of coarse Laves phases precipitated in the V-SLM specimen. These results are in keeping with the literature [13]. In the Ar-SLM specimen, the δ phase was formed by consuming the Nb in each fine Laves phase during heat treatment. Thus, the δ phase precipitated near Laves phase spots, and a large amount of the δ phase precipitated uniformly along the interdendritic regions. This led to brittle fractures along the interdendritic regions due to crack initiation along the δ phase during the creep test [11]. However, in the V-SLM specimen, the Laves phase was large and several δ phases precipitated around the Laves phases during heat treatment, even though the number of Laves phases was very small. Therefore, the δ phase precipitated around the Laves phase intensively, while the little δ phase precipitated in other regions. In other words, the δ phase precipitated discontinuously along the interdendritic regions. In addition, the number of δ phases was much smaller. Thus, the V-SLM specimen did not experience brittle fractures along the interdendritic region during the creep test. Although there were some voids and micro-cracks in the Laves phases, it was difficult for the cracks to propagate due to the lack of δ phase and its discontinuous distribution. Thus, the V-SLM specimen experienced ductile fracture, and the creep life was longer than that of the Ar-SLM specimen.

### 4.2. Microstructural Differences and Their Effects on Creep Properties

The microstructures of the Ar-SLM and V-SLM specimens were different. First, we consider the texture. This differed due to the difference in heat history during the SLM process. During the Ar-SLM process, the Ar gas circulated throughout the chamber [14], accelerating the cooling rate; in the vacuum environment, this effect was not a factor. Thus, the V-SLM specimens had a lower cooling rate during the process. In addition, the cooling rate during solidification is predicted by primary dendrite arm spacing. The relationship between primary dendrite arm spacing and the cooling rate is a linear function in a double logarithmic chart [15]. By using a graph from the literature [15], the cooling rates of both specimens could be calculated, as listed in Table 8. The spacing in the Ar-SLM specimen was variable, with values within 0.5–1.0 μm. The Ar-SLM specimen experienced a high cooling rate, and its value, $10^5$–$10^6$ K/s, was consistent with previous research [16]. Meanwhile, the V-SLM specimen experienced a much lower cooling rate. The low cooling rate led to a larger-grained texture, with grain orientation along the thermal gradient [17]. The V-SLM specimen had a larger grain size, and a large grain size is preferable for creep resistance [9]. Increased grain size can reduce the effect of grain boundary slide. Thus, the larger grain size of V-SLM specimen allowed for a lower creep strain rate and better creep life.

**Table 8.** Cooling rate and primary dendrite arm spacing of the as-built specimens.

| Specimen | Ar-SLM | V-SLM |
|---|---|---|
| Dendrite Arm Spacing (μm) | 0.5–1.0 | 10 |
| Cooling Rate (K/s) | $10^5$–$10^6$ | $6.0 \times 10^2$ |

In addition, the Ar-SLM specimen had various crystal orientations, while the V-SLM specimen had a preferable orientation. In Figure 3b, the size of the red and purple regions indicates a [001] orientation and a [$\bar{1}12$] orientation, respectively, according to a previous study [18]. That study found that these orientations have good creep resistance for Ni-based superalloys. Meanwhile, the Ar-SLM specimen did not have a preferable orientation, and it was predicted that the Ar-SLM specimen would have little creep resistance by preferred orientation. Thus, the preferable orientation of the V-SLM specimen led to a low creep strain rate.

### 4.3. Effects of Oxides on Microstructure and Creep Properties

IN718 is strengthened by $\gamma'$ and $\gamma''$, and the coprecipitation of both particles was confirmed in cast and wrought [19] and AM [20] IN718 materials. The $\gamma''$ has a volume fraction of about 13%, while the $\gamma'$ has a volume fraction of about 4% [21]. Thus, the $\gamma''$ phase is the main strengthening phase. However, it is metastable, and often transforms into the δ phase. There is a critical diameter of the $\gamma''$ phase for coherency loss [22]. However, the $\gamma'/\gamma''$ interface hinders the coarsening of the $\gamma''$ particle, due to Al enrichment at the interface [23]. Therefore, the $\gamma'/\gamma''$ interface improves the stability of the $\gamma''$ phase by forming coprecipitation. Although the $\gamma'$ phase is small in quantity, it is necessary for improving the stability of the $\gamma''$ phase.

The oxygen contents of the powders used for Ar-SLM and V-SLM were 190 ppm and 160 ppm, respectively. These values were too high, because the oxygen content of conventional cast materials is 20 ppm or less. In fact, the content of cast IN718 is only 7 ppm by mill sheet [24]. In the V-SLM specimen, the oxide content decreased by half during the SLM process (Table 6). This is the vacuum effect, in which contamination on the powder's surface is removed during laser radiation in a vacuum environment [24]. However, as the Ar-SLM had a large amount of oxygen, many oxides were observed, which were aluminum and titanium oxides (Figures 5 and 6a). As aluminum and titanium are constituent elements of the $\gamma'$ phase, it could be predicted that the Ar-SLM specimen would have a lower $\gamma'$ phase than the V-SLM specimen, and this might promote transformation of $\gamma''$ phase into δ phase. This is detrimental to creep properties. Meanwhile, the V-SLM specimen had fewer oxides

(Figure 6b). This means the V-SLM specimen had more $\gamma'$ phase, making it difficult for the $\gamma''$ phase to transform into $\delta$ phase due to coprecipitation with the $\gamma'$ phase. Thus, the V-SLM specimen showed better creep life, due to its promotion of $\gamma'$ phase precipitation.

**5. Conclusions**

The following conclusions can be drawn from this work:

(1)　The V-SLM showed a larger-grained texture. This included dendritic structures, and precipitation such as a Laves phase, which had a preferable orientation due to the lower cooling rate during the SLM process.

(2)　The V-SLM specimen had a small amount of $\delta$ phase and did not precipitate uniformly along interdendritic regions.

(3)　While the Ar-SLM specimen experienced brittle fractures due to crack propagation along the $\delta$ phase, the V-SLM specimen experienced ductile fractures due to voids in the Laves phase.

(4)　The V-SLM specimen showed better creep properties due to larger grain size and preferable grain orientation.

(5)　The V-SLM specimen had a lower Al/Ti-rich oxide density. This promoted coprecipitation of the $\gamma'$ and $\gamma''$ phases, and improved the stability of the $\gamma''$ phase. Thus, it was more difficult for it to transform to the $\delta$ phase, and the V-SLM specimen showed better creep life.

**Author Contributions:** Conceived and designed the experiments: T.N. (Toshiki Nagahari), K.K., S.N. and N.S. Performed the experiments: T.N. (Toshiki Nagahari) and T.N. (Taigi Nagoya). Wrote the paper: T.N. (Toshiki Nagahari) All authors have read and agreed to the published version of the manuscript.

**Funding:** This research was funded by the ALCA Program of the Japan Science and Technology Agency, JST (grant number JPMJAL1605).

**Conflicts of Interest:** The authors declare no conflict of interest.

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
