# Peer review of "Microstructure and Creep Properties of Ni-Base Superalloy IN718 Built up by Selective Laser Melting in a Vacuum Environment"

_metals, doi:10.3390/met10030362_

Round 1

Reviewer 1 Report

see attached file

Reviewer 2 Report

Here are some notes and comments to the article:

  • Page 2 rows 66 and 67 - please change the table numbers, parameters are in Table 2 not in Table 1 the same for chemical composition.
  • Microstructures in Fig.1 are too small (width of fig.1a is less than 35 mm - too small, not acceptable for serious reading), it is almost impossible to identify described features (dendrites orientation and so on) - could you mark it directly into the figures? Also for easier reading of microstructures I recommend to put Fig.1a-b for Ar-SLM and Fig. 1c-d for V-SLM.
  • Page 3 row 107 - please unify terminology, once the Laves phase is written with capital letter another laves phase with the small letter, correct it. You write that Laves phase ranged from 50-100nm. How it was measured?
  • Figure 2 - again, too small, enlarge it. No matter that is used higher magnification for structure detail, still not readable.
  • Figure 3a - columnar and equiaxed grains, could you mark those equiaxed areas because I see columnar grains only.....maybe if the figs have the bigger size I will see equiaxed grains.
  • Page 4 - rows 116-119 - your statement is, that V-SLM has larger grains. You observed specimens in a longitudinal direction, what about transverse cross-section. Did you measure actual grain size for both Ar-SLM and V-SLM specs? What is the result?
  • Figure 4 - too small for serious reading, enlarge it, please. And again why not to put figures of the same treatment next to each other (4a and 4b Ar-SLM; 4c-d V-SLM?)
  • Page 4 - row 131 - please write correct figure numbers ... "Compared to the δ phase in as-built specimen (Fig. 4d)?, those in HT specimen (Fig. 4d)? were substantially larger."
  • Page 5 - row 155 - Is the EDS mapping suitable method to determine the oxygen content in metals at all? Our experiences show that oxygen, carbon and hydrogen content in materials in not correctly showed by EDS mapping analysis.
  • Figs 5 and 6 - as I mentioned above, too small.
  • Figs 8 and 9 - too small and for Fig. 9 change arrangement of figures as mentioned in previous comments.
  • Page 8 - row 234 - Correct the cooling rate, it definitely was not 106 K/s.
